# Smart Nursing Wheelchairs: A New Trend in Assisted Care and the Future of Multifunctional Integration

**DOI:** 10.3390/biomimetics9080492

**Published:** 2024-08-14

**Authors:** Zhewen Zhang, Peng Xu, Chengjia Wu, Hongliu Yu

**Affiliations:** Rehabilitation Engineering and Technology Institute, University of Shanghai for Science and Technology, Shanghai 200093, China; zhangzw2021@yeah.net (Z.Z.); xp19921211@163.com (P.X.); fdasio5519690@gmail.com (C.W.)

**Keywords:** smart nursing wheelchair, assisted care, multiple-sensor fusion technology, human–machine interaction, multifunctional integration

## Abstract

As a significant technological innovation in the fields of medicine and geriatric care, smart care wheelchairs offer a novel approach to providing high-quality care services and improving the quality of care. The aim of this review article is to examine the development, applications and prospects of smart nursing wheelchairs, with particular emphasis on their assistive nursing functions, multiple-sensor fusion technology, and human–machine interaction interfaces. First, we describe the assistive functions of nursing wheelchairs, including position changing, transferring, bathing, and toileting, which significantly reduce the workload of nursing staff and improve the quality of care. Second, we summarized the existing multiple-sensor fusion technology for smart nursing wheelchairs, including LiDAR, RGB-D, ultrasonic sensors, etc. These technologies give wheelchairs autonomy and safety, better meeting patients’ needs. We also discussed the human–machine interaction interfaces of intelligent care wheelchairs, such as voice recognition, touch screens, and remote controls. These interfaces allow users to operate and control the wheelchair more easily, improving usability and maneuverability. Finally, we emphasized the importance of multifunctional-integrated care wheelchairs that integrate assistive care, navigation, and human–machine interaction functions into a comprehensive care solution for users. We are looking forward to the future and assume that smart nursing wheelchairs will play an increasingly important role in medicine and geriatric care. By integrating advanced technologies such as enhanced artificial intelligence, intelligent sensors, and remote monitoring, we expect to further improve patients’ quality of care and quality of life.

## 1. Introduction

With the continued growth of the world’s aging population, demand in the areas of medicine and geriatric care is increasing [1]. To meet these needs, intelligent technologies are becoming crucial in enhancing healthcare services and improving the quality of care. Recent decades have seen a focus on assistance robots, including smart nursing wheelchairs, aimed at improving the quality of life for older individuals [2,3,4]. While these innovations are often still under development, they represent a significant shift towards more integrated and versatile solutions in healthcare [5]. Furthermore, beyond aiding the elderly, smart nursing wheelchairs also provide vital assistance to children with severe disabilities. These devices offer adaptable features such as customizable support systems and programmable controls, which are essential for enhancing mobility, safety, and independence among younger users. This dual applicability underscores the versatility of smart nursing wheelchairs and expands their relevance, demonstrating their transformative potential across different age groups in the healthcare spectrum.

Smart nursing wheelchairs, as an example of such innovations, not only fulfill their primary role as mobility aids but also incorporate additional functionalities that are essential for comprehensive geriatric care. While ensuring mobility remains a fundamental priority, the integration of features such as posture adjustment, automated navigation, and health monitoring emphasizes the evolution of wheelchairs from mere mobility tools to complex healthcare devices. This multifunctional approach is necessary to address the diverse needs of the aging population efficiently and effectively. It reflects a broader trend towards enhancing the autonomy and safety of elderly patients, demonstrating how essential mobility support is augmented by these intelligent capabilities to offer a holistic care solution.

There are limitations when using traditional manual care wheelchairs in terms of limited maneuverability and dependence on external assistance. In addition, most nursing wheelchairs are single-function wheelchairs [6]. However, for disabled patients with different care needs [7,8,9,10], as shown in Figure 1, there is a growing demand for multifunctional-integrated wheelchairs that provide features such as posture adjustment, automatic lifting, assisted bathing and toileting, among others [11,12]. Disabled older people typically require assistance from caregivers with mobility and daily activities, which can cause some inconvenience and dependency in their lives. For disabled older adults, activities such as transferring from bed to chair, using the toilet, and bathing require significant physical effort from caregivers and may even result in muscle strain or injury [13,14,15,16,17]. Additionally, these processes pose significant safety concerns for disabled elderly people [18,19].

In this comprehensive review article, we look at the applications and importance of intelligent care wheelchairs in the areas of medical care and geriatric care. We will review the existing nursing functions, multiple-sensor fusion technology and human–machine interaction interfaces, highlight the need for multifunctional-integrated nursing wheelchairs, and provide insights into future development trends. Smart care wheelchairs represent a significant innovation in healthcare, offering unprecedented opportunities to improve the quality of medical and elderly care, reduce the workload of healthcare professionals, and improve the quality of life of patients.

## 2. Analysis of the Current Status of Nursing Wheelchair Research

The work in this section analyzes and evaluates the functionality of currently representative wheelchairs, some of which may only be in the laboratory research stage. Smart nursing wheelchairs are designed with the primary goal of enhancing care for disabled patients, addressing the challenges of caregiver shortages and the high workload of nursing staff. These wheelchairs are equipped with advanced technologies that facilitate essential daily activities, improve the autonomy of patients, and ensure safety. While the core focus of smart nursing wheelchairs is to provide comprehensive assistive care, they also incorporate certain features that support rehabilitation tasks. These features are intended not as a replacement for specialized rehabilitation equipment but as supplementary aids that can enhance the overall care and recovery process.

In the context of rehabilitation, nursing wheelchairs provide crucial support by enhancing patient mobility and facilitating certain aspects of physical recovery. While not primarily designed for comprehensive rehabilitation, smart nursing wheelchairs include features that indirectly support rehabilitation efforts. For example, the ability of these wheelchairs to adjust patient posture can help in mitigating the risks associated with prolonged immobility, such as pressure ulcers [20,21]. Additionally, certain models equipped with standing functions can assist in occasional standing exercises, which are beneficial for patients needing sporadic lower limb activity. These functionalities demonstrate how smart nursing wheelchairs complement traditional rehabilitation methods by integrating assistive features that cater to specific medical healthcare needs [22,23,24].

This section details how smart nursing wheelchairs integrate advanced engineering technologies, sensor capabilities, and biomedical insights to improve rehabilitation outcomes. As shown in Table 1, we analyze and classify existing wheelchairs based on three aspects: nursing function, intelligent navigation, and human–computer interaction. It can be observed that the majority of current nursing wheelchairs possess a singular function, and the integration of multifunctionality and intelligence is not truly realized.

For instance, wheelchairs with auxiliary nursing functions often lack intelligence, while those with intelligent operations frequently lack nursing functions. Despite the current situation where the combination of multifunctionality and intelligence lags behind, the trend of integrating intelligent features into multifunctional wheelchairs is becoming increasingly apparent with the development of artificial intelligence, intelligent sensors, and Internet of Things technology.

To comprehensively assess the current status and trends in smart nursing wheelchair technologies, this study employed a systematic literature search strategy. Searches were conducted in electronic databases including PubMed, IEEE Xplore, and Scopus. Keywords such as “smart nursing wheelchair”, “automated navigation”, and “human-machine interaction” were used to ensure coverage of various aspects of relevant technologies.

The inclusion criteria for the literature were as follows:Publication Date: Only studies published within the last five years were considered to ensure the timeliness and relevance of the technological information.Publication Type: Research published in peer-reviewed journals was selected to ensure the scientific integrity and authority of the referenced information.Relevance of Content: The literature must provide detailed descriptions of the technological applications or research development processes of nursing wheelchairs.

The exclusion criteria were as follows:Exclusion of conference abstracts and non-peer-reviewed literature, as these sources typically lack rigorous scientific validation.Studies not directly related to smart nursing wheelchair technologies were excluded to maintain the focus and accuracy of the research.Research published more than five years ago was excluded, unless it had significant historical impact on current technologies.

By employing these strict literature-selection methods, this study aims to provide a comprehensive overview of the development of smart nursing wheelchair technologies, offering a scientific basis for future technological innovations and applications. Through a systematic review of recent literature, we identified 232 publications related to nursing wheelchairs over the past five years. After rigorous screening based on relevance and scientific rigor, 128 publications met our inclusion criteria. The analysis of these studies revealed a focused interest in various technological aspects of smart nursing wheelchairs: 36 papers discussed nursing care functions, highlighting the essential care tasks facilitated by these advanced wheelchairs; 48 papers explored multiple-sensor fusion technology, indicating a robust trend towards enhancing wheelchair autonomy and safety; and 44 papers were dedicated to human–machine interaction (HMI) technologies, reflecting the importance of improving user interfaces for enhanced user interaction and control. This distribution underscores the multidisciplinary approach in research that combines engineering, user experience, and clinical needs to advance the field of smart nursing wheelchairs.

### 2.1. Supportive Care Functions

The supportive care functions of intelligent nursing wheelchairs play a central role in medical and geriatric care, aiming to enhance the quality of life for patients and alleviate the workload of medical professionals. However, in a survey of caregivers, due to the heavy workload, 47% of the 1163 caregivers surveyed reported suffering from strain injuries to various body parts caused by caregiving tasks [50,51,52]. These injuries not only affect their health but also contribute to a shortage of caregivers. To address this issue and reduce the workload on caregivers, smart nursing wheelchairs have been designed to automatically perform a series of assistance tasks such as mobility support, posture adjustment, and daily life care, thereby alleviating the physical burden on caregivers [53]. These wheelchairs not only enhance the quality of life for patients but also humanize caregiving tasks, effectively easing the manpower pressure in the caregiving industry.

As shown in Figure 2, we surveyed recent representative research on nursing robots with displacement functionality. As prolonged sitting in a wheelchair can lead to issues such as pressure ulcer [54], postural adjustment features are crucial considerations in the design of nursing wheelchairs. Nursing wheelchairs can mitigate the development and worsening of pressure injury by altering the user’s posture and redistributing pressure through adjustments in seat angle, height, and tilt. A newly developed multifunctional wheelchair called “ReChair”, which integrates mobility, posture adjustment, and rehabilitation functions, was introduced in a study by J. Wu et al. A preliminary evaluation of its performance was conducted [26]. The results indicate that for individuals with limited mobility, such as those with spinal cord injuries or the elderly, training in standing ability proves to be a beneficial aid in the recovery of physical functions in the elderly. The seemingly simple and frequent task of transferring from bed to chair or other living scenarios such as toileting and bathing can be very laborious for disabled elderly individuals [55]. Therefore, for assisted transfers, there are lift wheelchairs and floor transfer wheelchairs [56,57], back-carry transfer [58], and lateral transfer wheelchair [29]. Martin S. et al. compared a high suspension lifting device with a floor transfer device. It was shown that the suspended patient lifting method can effectively assist healthcare workers during patient transfers and reduce their fatigue [59]. However, these transfer devices are bulky and require ceiling modifications during installation, which limits their versatility. Mechanical arms have been added to the wheelchair, and the arms have been used to simulate the caregiver’s arms to help transfer the patient, which is effective but requires high cost of equipment and self-generated stability of the wheelchair [18,58,60,61,62]. Although this humanoid robot was expensive to develop and has not been extensively tested, it may have limitations in practical use depending on the application area and task. Panasonic Japan has developed an integrated bed and chair device that allows healthcare professionals to remotely control the combination and separation of motorized beds and recliners, thus preventing secondary injuries to patients during transfers [30]. To assist healthcare providers in transferring patients from a lying position in bed to a chair more effectively, a commonly employed solution is the use of a sliding transfer pad. Baptiste A et al. tested friction-reducing products on the market [63]. These friction-reducing devices have practical applications in clinical practice and can be effective in reducing friction during lateral patient transfers.

Addressing the aspect of assisted bathing, due to differences in lifestyle habits, there are variations in bathing preferences among elderly individuals in different countries. J. Zhang et al. classified current assistive-bathing robots [64]. Notably, Zlatintsi A et al. developed a comprehensive bathing system called I-Support, as illustrated in Figure 3. This system is equipped with various sensors, including an electric chair, a soft robotic arm, temperature sensors, and humidity sensors. The all-encompassing assistance and safety monitoring provided by this system can effectively aid the elderly in bathing, ensuring their safety throughout the bathing process [31].

On the other hand, Weibo Wang et al. introduced a bed-assistive bathing robot, utilizing a shower device called PAO developed by SAKImed in Shinjuku, Tokyo, Japan, which employs a slatted stretcher as a transfer mechanism [65]. Similarly, a series of bathing devices developed by the Japanese company OG Wellness ,located in Okayama, also facilitate the transfer of elderly individuals by employing a mobile chair mechanism that docks with the bathtub. These two methods share similarities and a common goal of reducing the burden on caregivers during patient transfers. From a safety perspective, the horizontal transfer method appears to be safer. Z He et al. theoretically proposed a new approach to integrate artificial intelligence technology with healthcare, utilizing ergonomic design principles to assist older adults with bathing tasks in a bathroom environment. The robot offers versatility, a user-friendly interface, and a high degree of safety. Experimental validation of the robot’s performance in a bathing environment provides guidance for future improvements and applications [32].

For assisted toileting, intelligent nursing wheelchairs can be equipped with toileting-assistive devices, including motorized seat lifting and seat tilting, allowing patients more autonomy in the toileting process while reducing the workload of caregivers [33]. Jinzhen Jin et al. developed an integrated electric wheelchair that combines functions such as standing, shifting, and assisting with toileting [66]. Similar research in this field has been conducted by Bostelman R et al., who developed a motorized wheelchair with a function of helping patients get up and move around, assisting in toileting, and transferring the seat, thereby making care more efficient [56]. However, these devices are bulky, expensive to produce, and have limited mobility. Ling Zhang et al. described the design of a smart wheelchair to assist older adults with daily toileting activities, such as standing up and sitting down while using the toilet [28]. Nevertheless, there are limitations in the size of the care wheelchair when transferring older adults from bed to wheelchair indoors, especially when frequent movement is required between places such as beds, toilets, and bathrooms. To overcome this challenge, McNamee wheels with zero turning radius are recommended as a mobility base for care wheelchairs [27,67]. This allows the wheelchair to navigate freely even in tight spaces. Although many assistive devices have been commercialized and are available on the market, their practical effectiveness often allows them to perform only specific nursing tasks. For instance, some wheelchairs are specially designed for posture adjustment, while others focus on aiding standing or transferring. This specificity in functionality, although catering to the needs of particular users, limits their broader application in more complex or varied daily care scenarios.

Recognizing these limitations is crucial for understanding the current level of technological development and application scope, and it highlights the challenges that future research and development efforts need to address. Future technological innovations should aim to develop more multifunctional and adaptable smart nursing wheelchairs, to more comprehensively meet the diverse needs of users and enhance their quality of life. Figure 4 illustrates the aforementioned relevant devices.

In Table 2, this table systematically organizes different assistive functions found in nursing wheelchairs, incorporating an evaluation of each function’s benefits, drawbacks, and specific criteria to determine its overall effectiveness and appropriateness.

In summary, the current nursing equipment face the following challenges:Limited Nursing Functions: Most nursing devices are designed to address specific nursing needs, necessitating the transfer of elderly individuals between devices when faced with different care requirements. This increases potential safety risks during the transfer process.Bulky Size: Home nursing devices often have a large footprint, requiring substantial operating space, which poses high spatial demands on home environments and limits the adaptability of the equipment.Space Modification Requirements: The use of nursing equipment often demands significant modifications to the user’s living space, resulting in additional costs for the user.Limited Maneuverability: The majority of electric nursing devices employ differential drive systems for chassis movement, leading to a large turning radius during mobility. This is not conducive to navigating through narrow indoor environments.

### 2.2. Multiple Sensor Fusion Technology

The multiple-sensor fusion technology of nursing wheelchairs uses sensors and navigation systems such as laser radar [68,69], cameras [70], ultrasonic sensors [71,72], etc., to achieve autonomous navigation, obstacle avoidance, and path planning. The methodology involves integrating these sensors using advanced algorithms to process and fuse data, creating a comprehensive environmental map.

In the early 1990s, Richard C. et al. introduced an adaptive assistance wheelchair system called NavChair [67]. This wheelchair is capable of sensing the indoor environment, including obstacles, furniture, and doors, via LiDAR and cameras while creating real-time maps of the environment. This endeavor integrates sensor technology, path planning, and user interface design, ultimately improving users’ independence and quality of life. During the COVID-19 pandemic, Zhang, Z et al. developed an automated navigation system [34]. This system uses AR marker technology and computer vision algorithms (YOLOv5) to detect objects in the environment. The design of this system reduced contact between health workers and the elderly during the pandemic, thereby reducing risks and conserving human resources. To increase the safety of wheelchair users, Haddad MJ et al. has developed an assistive decision system (PROMETHEE II) to help users control wheelchair direction and improve decision accuracy [35]. However, this study is still at the preliminary experimental stage due to the lack of comprehensive experimental validation and practical applications. In the area of environmental recognition, Christos et al. developed an indoor wheelchair navigation system based on RGB-D data fusion [36]. By integrating color images with depth information, this system improves environmental awareness as well as navigation accuracy and efficiency. This is potentially important for wheelchair users who require assistance with indoor navigation, improving their independence and mobility. In order to improve the adaptability of the wheelchair to different terrains, Wang C et al. conducted research on achieving stable autonomous navigation for robotic wheelchairs in ramped environments [37]. They used radar and tilt sensors to collect information about the ramps in the area and conducted a series of experiments to evaluate the wheelchair’s navigation performance under different slopes and environmental conditions. Similarly, Daniel et al. proposed a semantic navigation control method to help intelligent wheelchairs navigate through narrow passageways. The researchers used multiple sensors to capture environmental information, combining this information with a semantic data set to identify passageways, doors and other critical locations to support the wheelchair’s autonomous navigation. This technology helps plan safe paths and avoid obstacles and has the potential to improve wheelchair navigation capabilities in crowded or complex indoor environments [39]. Huang et al. conducted an overview and summary of the research and applications of indoor localization systems in recent years in the field of mobile robotics [73]. They analyzed the advantages and disadvantages of each localization method in terms of accuracy, cost, infrastructure change requirements, and robustness. They concluded that simultaneous localization and mapping (SLAM) based on laser radar and computer vision provides high precision (with an accuracy within 1–2 cm), good stability, and environment-independent properties without requiring changes to the environment compared to various existing localization methods. Therefore, it has the potential to become a significant trend in indoor localization and navigation in the future. In nursing wheelchairs, multiple-sensor fusion technology is used not only to avoid obstacles but also in wheelchair navigation for certain nursing tasks. Xie et al. proposed a sensor-fusion-based method for automatic bed–chair docking [38], which optimizes the bed–chair transfer process in medical care, thereby increasing efficiency and safety. This is of significant importance for improving the quality of medical care and work efficiency.

To better evaluate the effectiveness of the existing multiple-sensor fusion technology, we propose the following steps to validate its effectiveness:Simulation Testing: Simulated environments are created to test the initial functionality and performance of the sensor fusion algorithms.Field Testing: Real-world environments are used to test the nursing wheelchairs in practical scenarios, evaluating their ability to navigate and avoid obstacles.User Trials: Trials with actual users, including healthcare professionals and patients, are conducted to gather feedback on usability and effectiveness.Performance Metrics: Metrics such as accuracy, response time, and reliability are measured to ensure the technology meets the required standards.

Through the steps outlined above, we organized the multi-sensor technologies applied to wheelchairs and analyzed the effectiveness of each type of sensor, while also comparing the advantages and disadvantages of different sensor technologies in practical applications. The classification of sensors employed in the intelligent navigation systems of nursing wheelchairs, as outlined in Table 3, serves to underscore the diversity and specificity of technologies currently utilized to enhance the autonomy and safety of these assistive devices. Each sensor type—ranging from 2D and 3D LiDAR to ultrasonic, RGB-D cameras, GPS, IMU, and sonar sensors—is meticulously evaluated for its suitability and effectiveness within the context of wheelchair navigation. This table not only delineates the applications, advantages, and limitations of each sensor type but also suggests robust evaluation methods, such as field testing, simulation, and comparative analysis.

These evaluation methods are pivotal in determining the practical functionality and integration of sensors into the navigation systems. They assess aspects such as accuracy, environmental adaptability, user feedback, and computational demands, which are critical for ensuring the systems’ operational efficacy in real-world conditions. This structured approach to classification and methodological evaluation highlights the technological nuances and the need for tailored solutions in the development of intelligent navigation systems for nursing wheelchairs, aiming to improve patient mobility and caregiver efficiency.

In summary, the current multi-sensor fusion technology in nursing wheelchairs faces the following challenges:The environmental awareness capabilities of nursing wheelchairs are relatively limited in terms of intelligent navigation, making it difficult to accurately detect obstacles and the terrain in crowded indoor environments.Traditional care wheelchairs lack autonomy, often requiring manual control in common care scenarios such as bedside chair docking and toileting, resulting in inconvenience for both patients and caregivers.Future improvements in nursing wheelchairs should include the integration of advanced sensing technologies such as laser radar, cameras, and ultrasonic sensors to improve environmental awareness and improve the accuracy of obstacle detection. In addition, the development of smarter navigation and positioning algorithms should be promoted to give nursing wheelchairs more autonomy and navigation capabilities. This, in turn, would better support nurses in carrying out their nursing tasks, reduce nurses’ workload, and provide better nursing services to patients.

### 2.3. Human–Machine Interaction Functions

Modern nursing wheelchairs are equipped with a variety of human–machine interaction functions, such as voice recognition [40,42,53], touch screens [88], and remote controls [1,6,7,11,25,30]. These interfaces make it easier for users to operate and control the wheelchair, thereby increasing the user’s independence. Some care wheelchairs also have remote monitoring and medical record functions, allowing medical professionals to monitor the user’s health in real time.

The usability and operability of the human–machine interaction interfaces of intelligent care wheelchairs are of utmost importance. These interfaces allow patients or caregivers to communicate, control, and configure the wheelchair to adapt it to specific needs and preferences. Table 4 provides a detailed comparison of various human–machine interaction methods utilized in smart nursing wheelchairs. Each interaction mode is characterized by a specific description, highlighting its operational mechanism and contextual use. The table also lists the advantages and disadvantages of each method, offering insights into their practical applications and potential limitations in real-world settings. This comprehensive overview serves as a valuable resource for understanding how different interaction technologies can enhance the usability and functionality of nursing wheelchairs, thereby assisting in the selection of the most appropriate technologies based on specific user requirements and environmental conditions.

Akhil Raj et al. designed a smart wheelchair with touch screen and demonstrated the performance and accuracy of the Self-E system through extensive experiments [40]. Through patient testing and user experience, they showed that such touchscreen-equipped smart wheelchairs are better-suited to spinal cord injury patients, stroke patients, and older people with mobility problems than wheelchairs with traditional joysticks. For patients unable to operate the wheelchair with a joystick, Ahmed I et al. developed a voice-controlled wheelchair [41]. This wheelchair is equipped with a voice recognition system and also has intelligent navigation that ensures the safety of patients during the navigation process when using voice commands. To meet the needs of disabled elderly people, Sunny MSH and colleagues equipped wheelchairs with a six-degree-of-freedom robotic arm that assists patients in various daily tasks, such as picking, picking up objects, and opening doors, by controlling the movements of the robotic arm [42]. They also improved wheelchair control methods, thereby increasing the adaptability of wheelchair operation. This included using various parts of the user’s body to operate the wheelchair’s integrated assistance robot, such as movable fingers, the chin [95], the tongue [96], controlling the movement path of the wheelchair by detecting eye movements [42,92,93,94], and using the user’s facial expressions, such as smiling or blinking, to control the wheelchair [43]. Users can achieve directional control and stops using this technology, providing those with limited mobility a more intuitive and autonomous method of wheelchair control. In addition, there are common control methods based on head positioning and vector fields, where head positioning is used to control the direction and speed of the wheelchair, providing greater independence and autonomy [44]. Zhijun Li et al. proposed a navigation-controlled wheelchair with a shared human–machine control strategy using brain–machine interface (BMI) modes and autonomous control modes [45]. They presented a novel SSVEP based on a new BMI that utilizes two brain states to generate smooth polynomial paths and velocity profiles to satisfy the various dynamic constraints of the wheelchair. This improves movement safety and collision avoidance. Choi et al. has developed a common control system that combines BCI technology and laser radar perception to improve the intuitiveness and intelligence of wheelchair navigation. This is of great value for users who require specialized care, especially those who cannot use traditional methods of wheelchair control [89]. Similar studies involving the control of wheelchairs using brain–computer interface technology are also included [90,91,98,99]. Paul D et al. developed an intelligent system based on machine learning to identify the posture of wheelchair users [44]. This system uses a range of machine learning techniques, such as image processing and pattern recognition, to detect and recognize various user postures in real time, including sitting, lying, and standing postures. The system helps wheelchair users control their posture more effectively, improve their quality of life and reduce posture-related health problems. Cui et al. presented an intelligent wheelchair control system that utilizes multimodal sensing and various human–machine interaction methods to provide smarter, more convenient and user-friendly wheelchair navigation and control. They also emphasized integration with Internet of Things (IoT) technology [48]. In the field of health surveillance, A.Z.M. Tahmidul Kabir et al. developed a smart nursing wheelchair and built a mobile application [49]. The accompanying mobile application can monitor patients’ vital signs data in real time, including heart rate, body temperature, and blood oxygen saturation. The research also includes telemetry and remote control capabilities, allowing healthcare professionals to remotely monitor patient location, perform remote surgery, and provide remote support. The system records and analyzes vital signs, creates trend charts and alerts, helping healthcare professionals better understand the patient’s condition and recovery progress.

Building upon the discussion of current human–machine interaction (HMI) technologies in nursing wheelchairs, it becomes imperative to establish rigorous evaluation standards that ensure these systems are not only effective but also enhance user satisfaction and functionality. These evaluation criteria are designed to rigorously assess the efficacy of HMI systems, focusing on their integration within the complex environment of healthcare. In developing a comprehensive set of evaluation standards for HMI technologies, the criteria must address a broad spectrum of factors, including user-centric design, operational effectiveness, safety standards, feedback mechanisms, cognitive load, and maintenance demands. A user-centric design ensures that the HMI is accessible and adaptable to a wide range of users, including those with severe impairments, by incorporating features like voice control, adjustable feedback, and easily navigable interfaces. Such designs should also offer personalization options that can remember individual settings and preferences to enhance user comfort and ease of use. Operational effectiveness is another crucial criterion, emphasizing the reliability and efficiency of the HMI. The systems should demonstrate consistent performance across a variety of scenarios and be able to execute commands swiftly and accurately to avoid user frustration and potential safety hazards. Furthermore, HMIs should be equipped with robust error-handling capabilities that provide clear feedback for corrections and include emergency response features that are easily accessible in critical situations.

Feedback and communication are essential for building user confidence in the technology. Effective HMIs should employ multimodal interactions—combining visual, auditory, and tactile feedback—to accommodate users’ diverse needs and preferences. This approach not only improves the interaction but also ensures that the system can be used effectively by people with different sensory impairments. Cognitive load considerations are critical, as the system should not overwhelm the user with complex functionalities or require extensive learning periods. HMIs should maintain a balance between sophistication and simplicity, providing enough functionality in an intuitive manner that does not derail users’ focus from their primary activities. The ease of learning how to use the system and the availability of ongoing support and training also play vital roles in ensuring that users can fully benefit from the technology. Lastly, the durability and maintenance of HMIs are vital for long-term satisfaction and usability. The systems should be built to withstand regular use in various environments and be easy to maintain and update. This includes physical durability of components and the ease with which software updates can be implemented to keep the technology current and functioning optimally. By adhering to these comprehensive evaluation criteria, developers and healthcare providers can ensure that nursing wheelchairs are equipped with advanced HMI systems that are both innovative and aligned with the practical needs of users. Such standards not only drive the development of more effective technologies but also ensure that these innovations result in tangible improvements in the quality of life for wheelchair users.

In summary, although the current human–machine interaction functions of wheelchairs are diverse, they suffer from interface complexity and cumbersome operations. The main users of nursing wheelchairs are people with disabilities or semi-disabilities, especially older people. Simple and user-friendly human–machine interaction features, such as a large touchscreen interface, can help users with limited vision and hand coordination skills easily access features. By providing one-touch operation functions, the elderly can perform common tasks with ease, such as change sitting positions, adjust posture, use the toilet, and bathe, without the need for complex, multi-step processes. In addition, it is advisable to provide caregivers with convenient remote controls to operate the wheelchair when elderly people with disabilities need assistance.

## 3. The Importance Analysis of Integrated Functionality in Nursing Wheelchairs

Based on the technologies discussed above, we introduce the concept of multifunctional-integrated nursing wheelchairs and elucidate its significance. The main goal of this section is to explain how integrating various assistive functions—such as mobility support, postural adjustment, and automatic health monitoring—into a single device not only simplifies technological demands but also significantly enhances the quality of life for users. This part is crucial for understanding the practical application of advanced wheelchair designs in real-world care scenarios. By examining the combined effects of these functionalities, we demonstrate the transformative potential of smart nursing wheelchairs in the fields of medical and geriatric care, emphasizing their relevance to current healthcare challenges and the evolving needs of an aging population. Multifunctional integrated nursing wheelchairs represent a significant innovation in wheelchair design by combining supportive care features, intelligent navigation, and human–machine interaction capabilities. The importance of these integrated nursing wheelchairs is highlighted by their ability to meet complex care demands effectively and efficiently. The importance of multifunctional-integrated care wheelchairs is highlighted as follows:

Improved care efficiency: Multifunctional integrated care wheelchairs optimize various supportive care functions in a single device, such as B.Standing, transfers, bathing, and toileting. This significantly reduces nursing staff’s workload, saving time and resources and allowing them to focus more effectively on patients’ care needs.

Improved quality of life: These wheelchairs allow patients to participate in everyday activities such as standing, moving, and self-hygiene, promoting greater independence. This not only improves their quality of life, but also increases self-esteem and confidence and breaks down social and emotional barriers.

Increased quality of care: Intelligent navigation and sensor technologies enable multifunctional-integrated care wheelchairs to navigate and move safely, thereby avoiding collisions and injuries. This contributes to greater safety and accuracy in care and reduces the risk of accidents.

Improved user engagement: The human–machine interaction interfaces of multifunctional-integrated care wheelchairs allow users to easily control various functions via voice commands, touch screens, or remote control. This increases user engagement and autonomy, giving them the power to take control of their lives.

Adaptability to different needs: Different patients may have different care needs, and the configurability of multifunctional-integrated care wheelchairs allows them to adapt to different situations and individual differences. Adjustable settings allow wheelchairs to meet the specific needs and preferences of different patients.

Cost reduction: By integrating multiple functions into a single device, multifunctional-integrated care wheelchairs can reduce equipment purchase and maintenance costs as well as operating costs for care facilities.

In summary, multifunctional-integrated care wheelchairs not only increase care efficiency but also improve patients’ quality of life, increase care quality, increase user engagement, adapt to different needs, and reduce costs. This innovative care tool represents the future direction of medical and geriatric care, offering unprecedented opportunities to provide better care services, reduce caregiver burden, and improve patients’ quality of life. As technology continues to advance, the prospects for multifunctional-integrated nursing wheelchairs become even brighter.

## 4. Future Trends in Nursing Wheelchair Development

The future smart care wheelchairs are no longer merely traditional mobility devices; they come with various powerful assistive care features that can significantly reduce the workload of caregivers. These features include posture adjustments that facilitate easier standing for patients, promote muscle activity and blood circulation, and reduce the risk of pressure ulcer caused by prolonged immobility in a wheelchair. Additionally, nursing wheelchairs can provide assisted transfers to help patients move easily from bed or chair, thus reducing the workload of caregivers. Assisted bathing and toileting features can greatly enhance access to personal hygiene, especially for patients requiring specialized care. The integration of these care functions not only makes caregiving more efficient but also improves the patient’s comfort and self-esteem, helping caregivers alleviate occupational diseases [94,100]. Another key feature of smart nursing wheelchairs is their intelligent navigation capabilities. With the help of modern sensing technologies such as LIDAR, cameras, and ultrasonic sensors, smart care wheelchairs can recognize their surroundings, avoid obstacles, and autonomously plan safe paths. This allows patients to move more autonomously and no longer solely rely on caregivers for guidance [101]. Intelligent navigation also helps avoid accidental collisions and injuries, improving the safety of care. Nursing care functions and navigation technologies are complemented by human–computer interaction design. Users of care wheelchairs are usually mobility-impaired individuals who mostly face communication difficulties in terms of speech and gestures [25]. Therefore, the user-friendliness of the intelligent nursing wheelchair is crucial to ensure that the user can easily operate and control the functions of the wheelchair. In addition, the user interface needs to be highly customizable and adaptable to meet the needs of different patients. This human–computer interaction interface not only improves the operability of the intelligent nursing wheelchair but also enhances the user’s sense of involvement and autonomy.

Summarizing the trajectory of nursing wheelchair innovation, we envision a future where these devices are not merely functional but transformative, addressing comprehensive care needs through multifunctional integration. This evolution encompasses the integration of advanced features such as posture adjustment, enhanced mobility, and support for daily activities like bathing and toileting, thereby offering a holistic care solution. Moreover, future nursing wheelchairs will incorporate intelligent systems for autonomous navigation and obstacle avoidance, significantly reducing the reliance on caregivers. These wheelchairs will adapt to the specific physical needs of each user, with customizable seats and armrests tailored to optimize comfort and functionality. Furthermore, these advanced devices will be connected to cloud-based systems, enabling continuous remote monitoring and real-time data sharing. This connectivity will allow healthcare professionals to stay updated on patients’ conditions without being physically present. Additionally, the integration of diverse human–computer interaction interfaces—including sophisticated voice recognition, intuitive touchscreens, user-friendly head controls, and immersive virtual reality options—will greatly enhance user control and comfort. In essence, the next generation of intelligent nursing wheelchairs will focus on the following:Multi-functional integration: Integration of multiple assistive care functions such as multi-posture change, transfer, bathing, and toileting.Intelligent and automated: able to navigate autonomously, avoid obstacles automatically, and perform care tasks such as standing, lifting, bathing, and toileting according to the patient’s needs.Individualized design and customization: future nursing wheelchairs will have greater ability to be individually configured, with customized settings based on the patient’s physical condition, needs, and preferences.Remote monitoring and cloud-connected technology: the nursing wheelchair will integrate sensors for vital signs monitoring, such as heart rate, blood pressure, blood glucose, etc., to monitor the patient’s health in real time.Diversity of human–computer interfaces: future nursing wheelchairs will offer a variety of human–computer interfaces, which may include smarter voice recognition, more intuitive touchscreens, more convenient head controls, and virtual reality interfaces.

The nursing wheelchairs of the future feature multifunctional integration and intelligence, offering more comprehensive, efficient, and personalized solutions for medical and elderly care. We are also very interested in the understanding of this issue by artificial intelligence (AI). For this, we consulted ChatGPT 4.0. The question we asked was “ChatGPT, based on your imagination of the future development of nursing wheelchairs, how should we integrate multifunctional nursing wheelchairs? Please generate 7–8 images of future nursing wheelchairs based on your understanding”. Figure 5 shows our attempt to have ChatGPT draw its imagination of future care devices.

They will have smarter control systems that integrate artificial intelligence, machine learning, and autonomous decision-making capabilities, allowing the wheelchair to better understand the user’s needs and intentions. Artificial intelligence (AI) is rapidly transforming various aspects of healthcare, and its integration into nursing wheelchairs represents a significant innovation, offering enhanced independence and safety for users [69,102]. AI can be leveraged in multiple facets of wheelchair functionality, from navigation and user interface to health monitoring and emotional interaction.

Autonomous Navigation: One of the most promising applications of AI in nursing wheelchairs is in improving autonomous navigation. AI algorithms can process complex sensory data from the environment—such as obstacles, terrain, and movement patterns—to navigate safely and efficiently. This not only helps users with severe disabilities to move more freely but also reduces their cognitive load, allowing them to focus on other tasks.

Predictive Health Monitoring: AI can analyze data collected from integrated sensors monitoring vital signs like heart rate, blood pressure, and even blood glucose levels. By applying machine learning models, the system can identify patterns that may indicate a potential health issue before it becomes critical, providing alerts to both the user and healthcare providers. This proactive approach could be particularly beneficial for users with chronic conditions or those at risk of sudden health events.

Adaptive Controls and Personalization: AI can adapt the wheelchair’s controls based on the user’s physical ability and preferences, which can change over time. For example, the sensitivity of joystick controls can be adjusted automatically as the user’s motor skills evolve. Furthermore, AI can learn from user behavior to customize settings, such as preferred speeds and routes, enhancing the wheelchair’s usability and comfort.

Human–Machine Interaction Enhancements: Enhanced human–machine interaction is another area where AI can have a significant impact. Through natural language-processing and voice-recognition technologies, users can interact with their wheelchairs using simple voice commands. AI can also support gesture recognition, allowing users to control their wheelchair with intuitive gestures, which is especially useful for those who may have difficulty with traditional control systems.

Emotional Recognition: Integrating emotional recognition technology can enable the wheelchair to detect the user’s emotional state through voice tone analysis and facial expression recognition. This could allow the wheelchair to adjust its behavior in response to the user’s emotions, for example, stopping or slowing down when the user is distressed, thereby enhancing safety and providing emotional support.

Enhanced Safety Features: AI can also contribute to safety through anomaly detection in the wheelchair’s operation, such as detecting mechanical failures or unusual driving patterns that could lead to accidents. Immediate alerts can be sent to the user or a remote-monitoring center to take swift action.

These AI-driven features not only aim to improve the functional aspects of nursing wheelchairs but also focus on creating a more empathetic and responsive caregiving tool. By incorporating AI, nursing wheelchairs can become a pivotal element in modern healthcare, offering users greater autonomy, personalized care, and enhanced quality of life. The ultimate goal is to integrate these technologies seamlessly into the daily lives of individuals, ensuring that the wheelchairs are not just mobility aids but also proactive healthcare partners. These trends will significantly improve patients’ quality of life, reduce the workload of healthcare providers, and advance the nursing wheelchair field to meet the challenges of future healthcare needs.

## 5. Conclusions

This report summarizes the assisted-care functions, navigation and human–machine interaction capabilities of intelligent care wheelchairs, as well as the multifunctional integration trends for the future. Smart care wheelchairs not only provide essential mobility features, but also integrate various assisted-care functions, such as posture adjustments, transfers, bathing, and toileting. Their navigation and obstacle avoidance capabilities enable safe movement, while intelligent human–machine interaction interfaces allow users to easily control various wheelchair functions. Smart care wheelchairs hold significant potential to improve medical and geriatric care by improving patients’ quality of life and reducing caregiver workload. The use of this technology contributes to improved quality of care, greater patient autonomy, and the promotion of accessible healthcare and remote medical services.

In summary, intelligent care wheelchairs represent a crucial development direction for future medical and geriatric care. With continued technological advances, these wheelchairs offer more opportunities to provide better care services, reduce the burden on caregivers, and improve patients’ quality of life. We expect future research and innovation to further advance the development of intelligent care wheelchairs to meet changing healthcare needs.

## Figures and Tables

**Figure 1 biomimetics-09-00492-f001:**
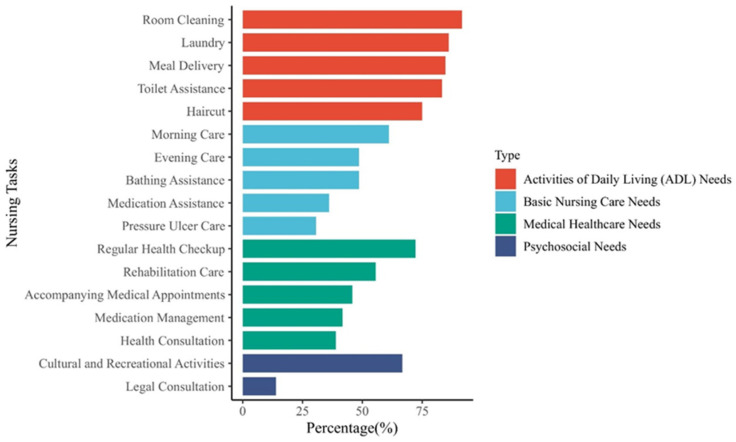
The required care services for disabled older people based on the dimensions of their living care needs, their primary care needs, their health needs, and the top five entries for each dimension, including items in the psychological well-being dimension [7,8,10].

**Figure 2 biomimetics-09-00492-f002:**
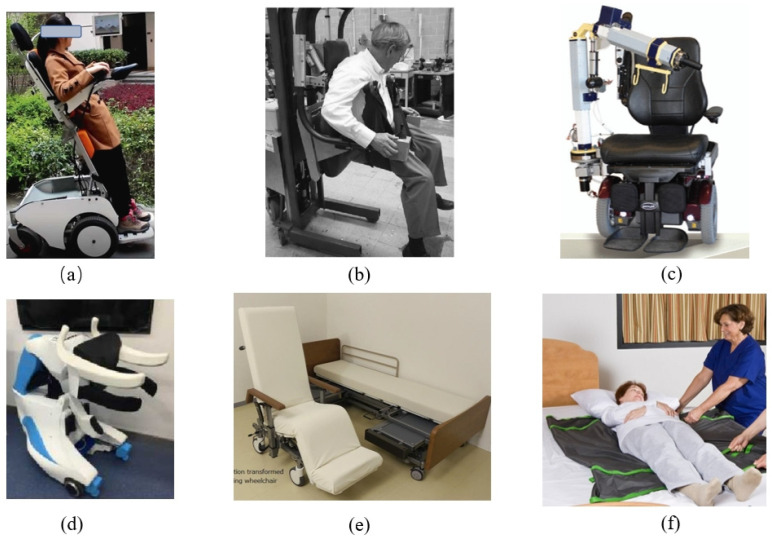
Typical assisted-lifting robots. (**a**) ReChair. (**b**) HLPR Chair. (**c**) The AgileLife Patient Transfer System and Strong Arm. (**d**) The Piggyback Transfer Robot. (**e**) Integration of an electric care bed and an electric reclining wheelchair. (**f**) Transferring the patient assisted by slipmat.

**Figure 3 biomimetics-09-00492-f003:**
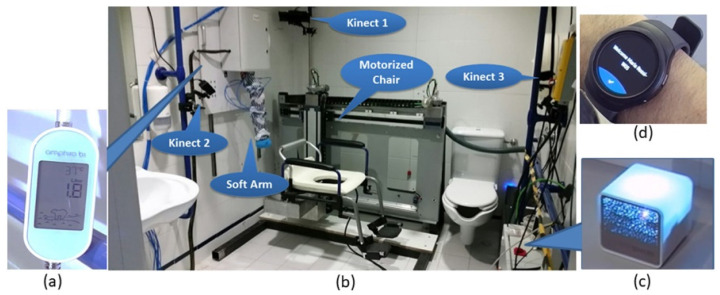
Installation of the I-Support system in clinical environment for experimental validation. The devices constituting the overall system are presented. (**a**) Amphiro b1 water flow and temperature sensor. (**b**) General aspect of the system showing the motorized chair, the soft robotic arm, and the installation of the Kinect sensors (for audio–gestural communication). (**c**) Air temperature, humidity, and illumination sensors by Cube Sensors. (**d**) Smartwatch for user identification and activity tracking.

**Figure 4 biomimetics-09-00492-f004:**
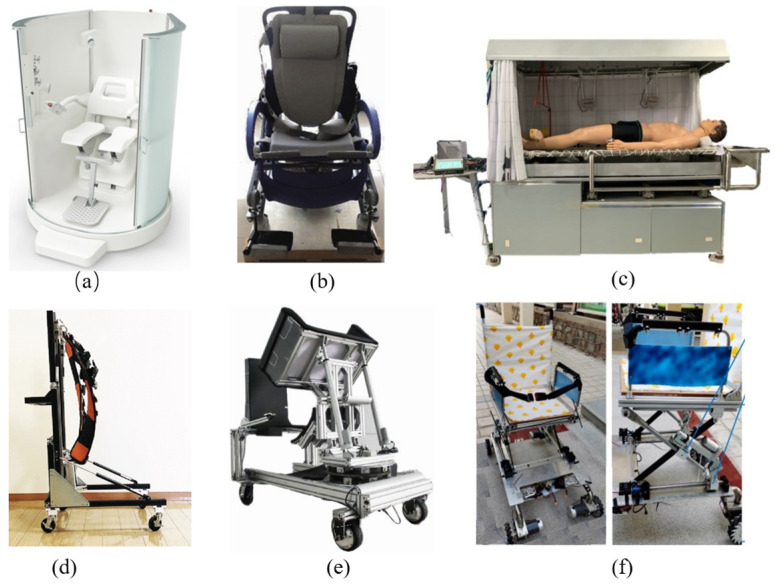
(**a**) Robotics Care’s om Poseidon (**b**) The multi-functional bathing robot. (**c**) The actual prototype of intelligent bath care system. (**d**) Assistive walker with passive sit-to-stand mechanism. (**e**) Self-reliance transfer support robot for home-based care. (**f**) Principle prototype of the intelligent toilet wheelchair.

**Figure 5 biomimetics-09-00492-f005:**
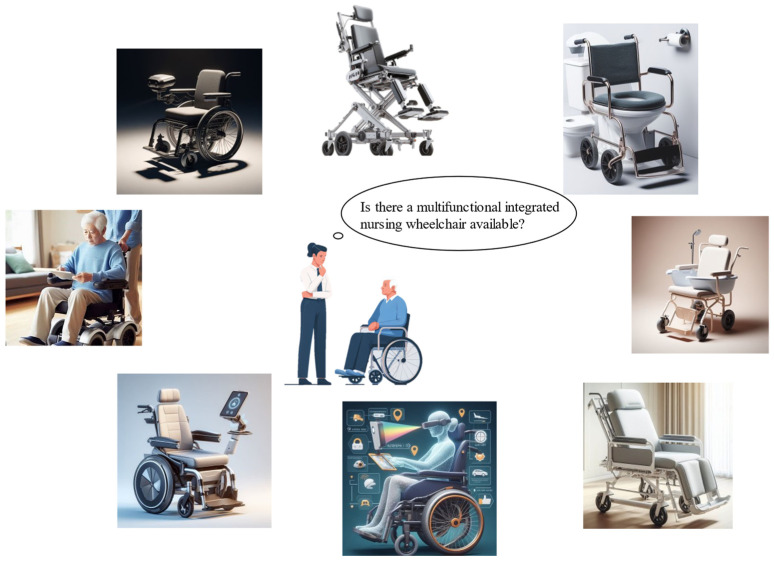
Future trends predicted by ChatGPT on multifunctional intelligent care wheelchairs.

**Table 1 biomimetics-09-00492-t001:** The cited nursing wheelchairs in this article are analyzed and categorized into three aspects: nursing care functions, multiple-sensor fusion technology, and human–machine interaction (HMI).

Paper	Nursing Care Functions	Multiple Sensor Fusion Technology	Human–Machine Interaction (HMI)
[10] Sang, L. (2019)	Transfer assist, lift assists, auxiliary toilet	None	Remote control
[25] Candiotti, J. (2019)	Assistive standing	None	Joystick control, navigation display screen
[26] Cao, W. (2021)	Assistive Standing, Multi-position adjustment	None	Joystick control, Voice control
[27] Thongpance, N. (2023)	N	None	Holonomic motion control
[28] Zhang, L. (2022)	Transfer assist, assistive standing	None	Remote control
[29] Koyama, S. (2022)	Bed-to-chair transfer assistance	None	Holonomic motion control
[30] Kume, Y. (2015)	Bed-to-chair transfer assistance	None	Remote control
[31] Zlatintsi, A. (2020)	Assistive bathing, transfer assist	None	Voice control, Gesture recognition control
[32] He, Z. (2019)	Assistive bathing	None	Intelligent AI recognition, Ergonomic design
[33] Shi, X. (2021)	Auxiliary toilet, assistive standing	None	Holonomic motion control
[34] Zhang, Z. (2022)	None	RGB-D	Navigation display screen, Joystick control
[35] Haddad, M.J. (2019)	None	Ultrasonic Sensor	Joystick control
[36] Sevastopoulos, C.(2023)	None	RGB-D	None
[37] Wang, C. (2020)	None	3D-LIDAR, IMU	None
[38] Xie, Y. (2022)	Bed-to-chair transfer assistance, multi-position adjustment	3D-LIDAR, RGB-D	None
[39] Correia, D. (2023)	None	LIDAR	Navigation display screen, Joystick control
[40] Megalingam, R.K. (2021)	None	LIDAR, IMU	Touchscreen navigation control
[41] Iskanderani, A.I. (2021)	None	Google Map	Voice control
[42] Sunny, M.S.H. (2021)	Robotic arm-assisted object retrieval	RGB	Eye-gaze control, Touchscreen navigation
[43] Rabhi, I. (2018)	None	Camera	Expression control
[44] Maciel, G.M. (2022)	None	RGB-D, IMU	head position control
[45,46] Li, Z. (2016)	None	Laser Sensor, EEG, RGB	BMI control
[47] Rosero-Montalvo, P.D. (2018)	Sitting posture monitoring	None	None
[48] Cui, J. (2022)	Vital signs monitoring, location detection, mobile environment detection	GPS, LIDAR, WIFI	Remote control, Gesture recognition control
[49] Kabir, A.T. (2023)	Vital signs monitoring, location detection	GPS, Infrared Sensor	Joystick control

**Table 2 biomimetics-09-00492-t002:** Integrated Evaluation of Assistive Functions in Nursing Wheelchairs.

Assistive Function	Description	Advantages	Disadvantages	Evaluation Criteria
Assistive Bathing [31,32,65]	Automated systems to aid patients in bathing activities.	Enhances independence; reduces caregiver strain.	High cost; complex maintenance requirements.	Cost-effectiveness: How the benefits align with costs. User impact: Effect on independence and caregiver reliance.
Bed-to-Chair Transfer [18,29,56,57,58]	Mechanisms facilitating transfers between beds and chairs.	Reduces physical exertion and risk of injuries.	Equipment cost and space requirements.	Operational ease: Simplicity of use. Safety: Risk of injuries to users.
Assistive Toileting [28,30,33,55,66]	Features facilitating the toileting process, such as automated seat adjustments.	Promotes patient dignity and independence.	Complexity in cleaning and maintenance.	Usability: Ease of cleaning and operation. Hygiene standards: Compliance with health and sanitation requirements.
Assistive Standing [26,27,53,54]	Support systems to aid users in standing up.	Supports rehabilitation and mobility.	Requires robust mechanical systems; potential safety risks.	Functionality: Support in daily activities. User safety: Ensuring the system is safe under all conditions.
Multi-Posture Adjustment [26,57,59]	Enables various seating adjustments to enhance comfort and health.	Prevents pressure ulcer; customizable to user needs.	Mechanism complexity leads to potential failures.	Reliability: Consistency and longevity of the mechanism. User comfort: Impact on user’s physical comfort and health.
Vital Signs Monitoring [31]	Sensors to monitor physiological parameters such as heart rate and temperature.	Allows continuous health monitoring; can alert to medical issues.	Increases cost; may raise privacy concerns.	Health impact: Effectiveness in improving patient monitoring. Privacy considerations: Handling of sensitive data.
Assistive Retrieval [60,61,62]	Robotic arms or similar mechanisms to help users retrieve objects.	Reduces dependency on caregivers for common tasks.	High initial and ongoing costs; complex mechanics.	Effectiveness: Ability to accurately perform intended tasks. Cost-efficiency: Economic viability given the benefits.

**Table 3 biomimetics-09-00492-t003:** Sensors commonly used in nursing equipment and their characteristics.

Sensor Type	Feature	Advantage/Disadvantage	Evaluation Methods
2D LiDAR [23,70,74,75]	Facilitates flat map creation; utilized for obstacle navigation	Relatively cost-effective; Limited by the absence of vertical information, leading to blind spots in intricate 3D environments.	Field Testing: Real-world performance in varied environments. Accuracy Assessment: Measurement of detection precision and range.
3D LiDAR [52,76,77,78,79]	Enables the generation of detailed 3D maps; adept at detecting and circumventing intricate obstacles	Provides a holistic environmental structure suitable for intricate navigation scenarios; Demands substantial computational resources for processing 3D.	Simulation Testing: Use in virtual environments to predict functionality. Integration Testing: Compatibility with other navigation systems.
RGB-D [36,70,80,81,82]	Offers visual information and delineates the 3D structure of the scene	Delivers high-resolution color imagery and depth data; Susceptible to lighting conditions, potential performance degradation in low-light or non-uniform lighting scenarios	Operational Testing: Evaluation in controlled environments to measure reliability and range. User Feedback: Collection of practical usage data from operators.
Ultrasonic Sensor [71,72]	Employed for short-range obstacle detection and avoidance	Economical solution; unaffected by lighting conditions; limited precision.	Comparative Analysis: Benchmark against other sensor types for object recognition accuracy. Environmental Testing: Assess performance across different lighting conditions.
Infrared Sensor [83,84]	Utilized for detecting distance, temperature, and related parameters	Cost-effective; Influenced by lighting and environmental temperature.	Precision Mapping: Evaluation of positioning accuracy in diverse geographic settings. Durability Testing: Long-term reliability and signal consistency.
GPS [85,86]	Designed for outdoor large-scale navigation and positioning	Exhibits high precision in outdoor navigation; Inapplicable for indoor navigation.	Precision Mapping: Evaluation of positioning accuracy in diverse geographic settings. Durability Testing: Long-term reliability and signal consistency.
IMU	Employed for attitude estimation, motion control, and navigation	High-frequency data updates; adaptable to dynamic scenarios; cumulative errors over time result in drift.	performance Metrics: Analysis of drift and correction mechanisms. Sensor Fusion Analysis: Effectiveness in integration with other technologies like GPS or LiDAR.
Sonar Sensor [68,87,88]	Utilized for distance measurement, obstacle detection, and positioning	Utilized for distance measurement, obstacle detection, and positioning.	Range Testing: Evaluate effective operational range and sensitivity. Robustness Analysis: Assess performance against environmental variables like humidity or temperature.

**Table 4 biomimetics-09-00492-t004:** Human–machine interaction methods in smart wheelchairs: advantages and disadvantages.

Interaction Mode	Description	Advantages	Disadvantages
Touchscreen [40,88]	Interface that allows users to interact with the system via touch inputs.	Intuitive and user-friendly; suitable for routine operations.	May be difficult to use under direct sunlight or in brightly lit conditions.
Voice Recognition System [40,42,89]	Technology that allows the wheelchair to be controlled through spoken commands.	Allows hands-free operation; convenient for voice commands.	May be affected by ambient noise; sensitive to accents and speech variations.
Remote Control [6,7,11,25,30]	A device or system that enables the wheelchair to be controlled from a distance.	Enables users or caregivers to control the wheelchair remotely.	Requires carrying a remote; risk of signal interference.
Brain–Machine Interface (BMI) [45,48,89,90,91]	An interface that translates neuronal information into commands capable of controlling the wheelchair.	Suitable for users with severe mobility restrictions.	Requires specific training; high cost; practicality depends on technological advancement.
Body Part Control (e.g., gestures, foot) [47]	Systems that allow wheelchair control using different body parts like feet or other gestures.	Allows users with limited hand mobility to control using other body parts.	Requires some physical coordination ability; may not be suitable for all users.
Eye Movement Control [42,92,93,94]	Technology that tracks eye movements to control the wheelchair.	Provides an efficient control method for users with extreme mobility limitations.	Requires high precision technology; may need time for users to adapt.
Facial Expression Control [43]	Systems that use facial expression recognition to control the wheelchair.	Controls through user facial expressions, so no need for hand or voice operation.	Needs highly sensitive sensors and advanced algorithms.
Head Movement Control [44,95,96]	Interfaces that use the direction and angle of the head for wheelchair control.	Allows users to control direction and speed by moving their head.	May not be suitable for users with neck injuries or conditions.
Health-Monitoring Interaction [97]	Systems that integrate health monitoring sensors to manage and respond to physiological data.	Monitors vital parameters like heart rate, ensuring safety.	Requires ongoing data processing and privacy protection measures.

## Data Availability

All data generated or analyzed during this study are included in this published article.

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
