# Peer review of "Smart Nursing Wheelchairs: A New Trend in Assisted Care and the Future of Multifunctional Integration"

_biomimetics, 2024, doi:10.3390/biomimetics9080492_

Round 1

Reviewer 1 Report

Comments and Suggestions for Authors

The article is well written and valuable.  Please consider the following suggestive enhancements:

1. Enhance the resolution and dimensions of Figure 4 to ensure clarity and better visual impact.

2. While the layout and alignment of Tables 1 to 3 are consistent, there appears to be a discrepancy in the logic presented in the final columns of Tables 1 and 2, compared to Table 3 (with respective raw VS without). This might result in some confusing.

3. The phrases "Operational ease: ..." and "Safety: ..." are duplicated in Table 2 in varying formats. To avoid redundancy, **consolidate these terms** into a single, coherent statement.

4. In addition to the elderly, the smart nursing wheelchair also offers significant assistance to children with severe disabilities. Highlighting this application within the main text would underscore the device's versatility and broad relevance.

5. One of the critical challenges is “integrating multiple functions into a single, cohesive unit”. Addressing this challenge within the article will provide a more holistic view of the smart nursing wheelchair's potential and limitations.

Reviewer 2 Report

Comments and Suggestions for Authors

Dear Author(s),

I have gone through the submitted Manuscript Ref. No. BIOMIMETICS-3099225 entitled “Smart Nursing Wheelchairs: A New Trend in Assisted Care and the Future of Multifunctional Integration” and I have the following observations, which I hope it will help in adding value to the submitted manuscript:

1.     In my opinion, the Manuscript Ref. No. BIOMIMETICS-3099225 could be classified as a “review” paper. In this case, I consider the number of references should be considerably increased.

2.     Figure 1 – “Rehabilitation” is an extensive term, which covers many directions, according to the definition provided by the World Health Organization. I propose to the authors to offer more detailed information – to provide a connection between “rehabilitation” directions and their findings about the “smart nursing wheelchairs” state of art.

3.     Section 2.1, line 82

I think it would be helpful to provide some references (studies) to emphasize the amplitude for the shortage of healthcare workers. Maybe it will be a good idea to consult materials like

-        https://doi.org/10.3390/healthcare11131887

4.     Section 2.1, line 84

To provide readers with a good image about the amplitude of the research interest in developing intelligent nursing wheelchairs, I recommend sustaining your affirmation with some quantifiers.  

5.     Section 2.2

As stated in the paper, the research interest on smart wheelchairs dates way back in time. The first reported research on smart wheelchairs that approached the navigation aspects and also a sensor fusion technology dates back to 1986 when Madarasz used a digital camera and sonar to provide autonomy to the wheelchair movement. NavChair versions used several sonars as sensors for the obstacles. The research in this field spans over almost 40 years, and it includes a large number of publications, thus I recommend extending the number of references included in the paper. I think it may be helpful to consult publications like:

-        https://doi.org/10.1007/978-3-319-05951-8_52

-        https://doi.org/10.1109/ICSTCC.2019.8886158

6.      Section 2.2, line 198

Since the authors looks to evaluate the effectiveness of the existing multiple sensor fusion technology, I propose the use the MDPI Biomimetics recommendations for research sections (https://www.mdpi.com/journal/biomimetics/instructions#preparation): and to organize the information in subsections like “Materials and Methods”, “Results”, “Discussion”, “Conclusions”. I.e., lines 198 – 2008 could be reformulate as a part of a “Materials and Methods” subsection. Also, to improve understanding, this subsection should elaborate on the methods used to derive the information presented in “Table 3”.

7.     Section 2.2, lines 270 – 277

In my opinion, those lines are an enumeration, thus I consider that the authors should use eighter the Bulleted list or Numbered list format, from the Biomimetics paper template.

8.     I have similar recommendations to the ones previously stated, for Section 2.3. In addition, I consider it opportune to produce a Table data, similar to Table 2 for Section 2.1 and Table 3 for Section 2.2, in order to keep a homogenous presentation of information.

9.     Please emphasize the importance of Section 3, in the paper context – what is its purpose, objective, relevance?

10.  Section 4, lines 492 – 522

There it looks to be an enumeration to something, but I couldn’t identify it’s start and purpose.

11.  Section 4. The authors indicate the use of artificial intelligence instruments. Please consult the “Authorship and the Use of AI or AI-Assisted Technologies” paragraph on the Authorship section, on the “MDPI Research and Publication Ethics” webpage  https://www.mdpi.com/ethics#_bookmark3  , in order to assure the paper compliance with the MDPI provisions.

Reviewer 3 Report

Comments and Suggestions for Authors

This is a very interesting compilation/discussion of the current literature in this area.  I am not familiar with the term "nursing care wheelchair" but it is adequately described for understanding in the article.   

I am not sure this article is appropriately described as a "review" article in the traditional sense.  It does seem to be sort of a "scoping review" of the technologies that have been described in the literature.  It does not seem to contain the scientific elements of a traditional review article as it does not have any analysis of research methodology, etc. it merely presents some descriptive information about the technologies - which is interesting, but not what I think of when I think of a "review" article. 

I would like to see some information about how the literature search was performed and what the inclusion/exclusion criteria were for relevant literature to present.  there is not any indication of how these particular articles were selected and how the authors discovered them. 

in line 87 - the term "bedsore" is used and should be replaced with the more appropriate term "pressure injury" or "pressure ulcer" according to the conventions of the EPUAP or the NPIAP in the US.  In several other places the term "pressure sore" is also used  and should be replaced with either pressure injury or pressure ulcer. 

It would be interesting to have some additional discussion of the potential for commercially available devices in the areas discussed - for example, the technologies around various interface devices are well developed and commercially available already, as are technologies that facilitate posture changes (i.e. tilt, recline, standing features), that can already be obtained on  a variety of commercially available products, vs. those that are very "futuristic" such as robotic technologies or technologies that assist with activities like transfers or bathing.  All technologies described are presented in a similar manner with not much indication about the potential availability or how well developed the technologies are. 

in line 368 - the phrase "detrail users" is indicated - do you mean "derail users" - just a typographical error, unless I am not understanding the meaning.

It would be nice to see some discussion of wheelchairs as mobility aids - as they are currently intended. while it is interesting to think about the potential contributions of various technologies to other ADL or care assistance, it is important not to lose sight of the fact that wheelchairs are first and foremost mobility devices, so I would like to see some discussion of this early in the manuscript.  

an interesting concept and I did find the manuscript enjoyable to read.

Round 2

Reviewer 2 Report

Comments and Suggestions for Authors

Dear Author(s),

I have gone through the revised Manuscript Ref. No. BIOMIMETICS-3099225 entitled “Smart Nursing Wheelchairs: A New Trend in Assisted Care and the Future of Multifunctional Integration” and I have the following comments regarding the revised version, which I hope it will help in adding value to the submitted manuscript:

1.     Comment / Response 1 – The authors revised the manuscript and added 49 additional references. In my opinion, the authors fulfilled the initial review comment.

2.     Comment / Response 2 – The purpose of the initial comment was to create an image, for the readers, about the share of “smart nursing wheelchairs” inside the Rehabilitation care direction, related to “Medical healthcare needs” type of “Nursing tasks”, as depicted in Figure 1. The authors provided an emphasis between rehabilitation applications and the functionality of smart nursing wheelchairs, but without providing additional information related to the anterior-mentioned context. In my opinion, minor improvements can be made.

3.     Comment / Response 3 – The authors revised the manuscript and included a discussion on the shortage of healthcare workers, including the causes of this issue. In my opinion, the authors fulfilled the initial review comment.

4.     Comment / Response 4 – Considering the statement targeted by Comment 4, referring to Section 2.1, line 84 of the initial version of the Manuscript (“the development of intelligent nursing wheelchairs has garnered significant attention and research in recent years”), and given the “review” character of the Manuscript, I was expecting some statistical quantifiers / analysis – like number of papers published per each of the last five years, and maybe a categorization of number of papers tackling “Nursing Care Functions” / “Multiple Sensor Fusion Technology” / “Human-machine Interaction (HMI)”, or maybe another metric of characterization like the type of transfer type assist, or type of sensor, etc., which then to lead to the data contained in Table 1, and maybe even expanding it with future papers. I couldn’t identify such quantifiers in the revised Section 2.1. I consider that improvements can be made here.

5.     Comment / Response 5 – The authors revised the manuscript and expanded the references throughout the document. In my opinion, the authors fulfilled the initial review comment.

6.     Comment / Response 6 – The authors opted to maintain the current structure to provide a streamlined narrative that aligns closely with the paper's theme without overcomplicating the presentation. Nevertheless, I have noted that the authors consolidated the Section 2 – the introductory paragraphs, with extensive criteria set for selecting the relevant literature in the Manuscript context, which in my opinion is equivalent with specifying the equivalent “Materials and Methods”.
The authors also revised the Manuscript to present the methods used to evaluate the sensor technologies presented in Table 3

Also, by considering the authors arguments provided for the Comment no. 6, I think that the Comment is addressed sufficiently.

7.     Comment / Response 7 – The authors revised the Manuscript accordingly, fulfilling the initial review comment.

8.     Comment / Response 8 – The authors revised the manuscript and added a new table, like Table 2 and Table 3, to Section 2.3 of the manuscript. In my opinion, the authors fulfilled the initial review comment.

9.     Comment / Response 9 – The authors revised the manuscript and explicitly describe the connection between this section and the overall manuscript. Nevertheless, I believe there is a small material error on the line 463, but with a big impact on the Manuscript.
In the Response 9, the authors state
“we introduce a new concept based on existing nursing wheelchair technologies—the multifunctional integrated nursing wheelchair”.
But on the Manuscript, line 463, the text reads “we propose a new solution for nursing wheelchairs and elucidate its significance”, thus leading the reader to expect a proposed hardware and software for a nursing wheelchair. Also, taking in consideration the same Response 9, I think a rephrase on the line 473 should be taken in consideration by the authors, to align the Response 9 with the Manuscript content – “Multifunctional integrated care wheelchairs” vs “multifunctional integrated nursing wheelchair”.
In my opinion, minor improvements can be made.

10.  Comment / Response 10 – Initial comment: “Section 4, lines 492 – 522. There it looks to be an enumeration to something, but I couldn’t identify it’s start and purpose.” I was referring to the paragraph:

“(…). AI can be leveraged in multiple facets of wheelchair functionality, from navigation and user interface to health monitoring and emotional interaction.”

Then, on the next lines, one could read the following paragraphs:

Autonomous Navigation: (…)”

Predictive Health Monitoring: (…)”

(…)
Which looks like an enumeration of something.

In my opinion, maybe because of different line numbers displayed in the reviewer document vs. the authors document, the initial comment wasn’t handled adequately.

11.  Comment / Response 11 – The mentioned resource, in “MDPI Research and Publication Ethics”, states:

-        In situations where AI or AI-assisted tools have been used in the preparation of a manuscript, this must be appropriately declared with sufficient details at submission via the cover letter. Furthermore, authors are required to be transparent about the use of these tools and disclose details of how the AI tool was used within the “Materials and Methods” section,” – I believe that is in the authors best interest to specify a better borderline between their work and the ChatGPT vision. In my opinion, a reader could interpret that the paragraphs mentioned in Comment 10 could be AI generated.   

-        in addition to providing the AI tool’s product details within the “Acknowledgments” section.” I couldn’t identify such an information in the “Acknowledgments” section.
In my opinion, minor improvements can be made.

12.  Comment 12 – Also, since the reviewing / revising process makes a Manuscript “alive”, by reading the actual version of the manuscript, I have a new suggestion: I noticed that the Table 1 contained references to published papers since the initial version, and now the Table 3 also includes references to published papers. This is a good idea, which I consider that should also be applied to Table 2 and Table 4.

Author Response

Comments 1: Comment / Response 2 – The purpose of the initial comment was to create an image, for the readers, about the share of “smart nursing wheelchairs” inside the Rehabilitation care direction, related to “Medical healthcare needs” type of “Nursing tasks”, as depicted in Figure 1. The authors provided an emphasis between rehabilitation applications and the functionality of smart nursing wheelchairs, but without providing additional information related to the anterior-mentioned context. In my opinion, minor improvements can be made.

Response 1: Thank you for your detailed review and valuable comments regarding our manuscript. In response to your suggestion to provide a more detailed discussion of the role of smart nursing wheelchairs within the rehabilitation sector, we have made specific amendments and additions to the text.

To address your comments, we have included a new section that defines and contextualizes the primary design purposes of smart nursing wheelchairs, which are aimed at enhancing care efficiency and addressing the shortage of nursing staff and their workload. Although these wheelchairs primarily focus on assistive care, we have also detailed how they support specific rehabilitation tasks. This includes features like multi-posture adjustment capabilities, which help long-term bedridden patients change positions to mitigate the risk of pressure ulcers from prolonged immobility, and standing functions that assist with occasional standing exercises beneficial for patients requiring lower limb rehabilitation.

We hope these revisions clearly demonstrate the dual role of smart nursing wheelchairs in medical care and rehabilitation, emphasizing their supportive functions in the rehabilitation field without replacing specialized rehabilitation equipment. This narrative aims to provide readers with a comprehensive understanding of how smart nursing wheelchairs not only facilitate daily care but also augment the rehabilitation process.

Thank you once again for your meticulous review and helpful suggestions. We look forward to your further guidance and feedback.

Comments 2: Comment / Response 4 – Considering the statement targeted by Comment 4, referring to Section 2.1, line 84 of the initial version of the Manuscript (“the development of intelligent nursing wheelchairs has garnered significant attention and research in recent years”), and given the “review” character of the Manuscript, I was expecting some statistical quantifiers / analysis – like number of papers published per each of the last five years, and maybe a categorization of number of papers tackling “Nursing Care Functions” / “Multiple Sensor Fusion Technology” / “Human-machine Interaction (HMI)”, or maybe another metric of characterization like the type of transfer type assist, or type of sensor, etc., which then to lead to the data contained in Table 1, and maybe even expanding it with future papers. I couldn’t identify such quantifiers in the revised Section 2.1. I consider that improvements can be made here.

Response 2: Thank you for your constructive feedback. In response, we have incorporated the requested statistical data into the manuscript, now detailed in lines 131-145 of Section 2. This addition provides a comprehensive analysis of publications over the past five years, categorized by their focus on Nursing Care Functions, Multiple Sensor Fusion Technology, and Human-Machine Interaction. We hope this update aligns with your expectations and enhances the manuscript’s rigor.

Comments 3: Comment / Response 9 – The authors revised the manuscript and explicitly describe the connection between this section and the overall manuscript. Nevertheless, I believe there is a small material error on the line 463, but with a big impact on the Manuscript.

In the Response 9, the authors state “we introduce a new concept based on existing nursing wheelchair technologies—the multifunctional integrated nursing wheelchair”.

But on the Manuscript, line 463, the text reads “we propose a new solution for nursing wheelchairs and elucidate its significance”, thus leading the reader to expect a proposed hardware and software for a nursing wheelchair. Also, taking in consideration the same Response 9, I think a rephrase on the line 473 should be taken in consideration by the authors, to align the Response 9 with the Manuscript content – “Multifunctional integrated care wheelchairs” vs “multifunctional integrated nursing wheelchair”. In my opinion, minor improvements can be made.

Response 3: Thank you for your insightful suggestions regarding the consistency of terminology and clarity in our manuscript. We have carefully reviewed your comments and have revised the manuscript to better reflect the conceptual nature of our proposal for multifunctional integrated nursing wheelchairs.

Specifically, we have adjusted the text to replace “we propose a new solution for nursing wheelchairs” with “we introduce the concept of multifunctional integrated nursing wheelchairs” in line 493. This change clarifies that our focus is on a conceptual innovation integrating various assistive functionalities into a unified wheelchair system, rather than proposing specific hardware and software solutions.

Furthermore, we have standardized the terminology throughout the manuscript to consistently use “multifunctional integrated nursing wheelchairs.” This revision ensures that our description aligns with the conceptual focus and enhances the manuscript’s coherence regarding the transformative potential of these wheelchairs in medical and geriatric care.

We hope these modifications address your concerns and improve the clarity and consistency of our manuscript. We appreciate your guidance and look forward to your further feedback.

Comments 4: Comment / Response 10 – Initial comment: “Section 4, lines 492 – 522. There it looks to be an enumeration to something, but I couldn’t identify it’s start and purpose.” I was referring to the paragraph:

“(…). AI can be leveraged in multiple facets of wheelchair functionality, from navigation and user interface to health monitoring and emotional interaction.”

Then, on the next lines, one could read the following paragraphs:

“Autonomous Navigation: (…)”

“Predictive Health Monitoring: (…)”

(…)

Which looks like an enumeration of something.

In my opinion, maybe because of different line numbers displayed in the reviewer document vs. the authors document, the initial comment wasn’t handled adequately.

Response 4: Thank you for your valuable feedback on our manuscript. We have revised the section in question to clearly articulate the start and purpose of the enumeration as you suggested. The revised section now succinctly summarizes and forecasts the future trends of nursing wheelchairs following our discussion of current technologies.

We have included an introduction at the beginning of this section (lines 562-592), which sets the stage for the enumeration and outlines its intent. This introduction provides a clear and structured overview of anticipated advancements in wheelchair technology, thereby enhancing the reader's understanding of this crucial aspect of our review.

We believe these revisions address your concerns and greatly improve the clarity and effectiveness of our manuscript. We appreciate your guidance and look forward to your further comments.

Comments 5: Regarding the issue of artificial intelligence usage in the article.

Response 5: Thank you for pointing out the need for clarity regarding the use of CHATGPT in our manuscript. We appreciate your attention to detail and agree that a more detailed explanation would enhance the reader’s understanding of how we employed this tool.

In response to your comment, we have added a specific section that describes how we utilized ChatGPT to explore and visualize the future of nursing wheelchairs. As you suggested, the new content provides a clear example of our interaction with ChatGPT, detailing the exact query we posed: “ChatGPT, based on your imagination of the future development of nursing wheelchairs, how should we integrate multifunctional nursing wheelchairs? Please generate 7-8 images of future nursing wheelchairs based on your understanding.”

This addition, found in lines 598-602 of the revised manuscript, aims to transparently illustrate how ChatGPT contributed to our conceptualization of future technological integrations in nursing wheelchairs. We hope this revision satisfactorily addresses your concern and enriches the manuscript.

Thank you once again for your constructive feedback. We look forward to your further comments.

Comments 6:Also, since the reviewing / revising process makes a Manuscript “alive”, by reading the actual version of the manuscript, I have a new suggestion: I noticed that the Table 1 contained references to published papers since the initial version, and now the Table 3 also includes references to published papers. This is a good idea, which I consider that should also be applied to Table 2 and Table 4.

Response 6: Thank you for your continued engagement with our manuscript and for your insightful suggestion to enhance the consistency and informational depth of our tables.

We appreciate your positive feedback on the inclusion of references in Tables 1 and 3. In line with your suggestion, we have now revised Tables 2 and 4 to include references to published papers. This adjustment not only enriches the tables by providing direct sources for the data presented but also aligns all tables in the manuscript with a consistent format, thereby improving the overall coherence and scholarly rigor of our work.

We believe these changes will make it easier for readers to verify and further explore the data and analyses we have presented. Thank you once again for your constructive feedback, which has undoubtedly improved the quality of our manuscript.
